# Scoping Review of Peer-Led Support for People Bereaved by Suicide

**DOI:** 10.3390/ijerph19063485

**Published:** 2022-03-15

**Authors:** Agnes Higgins, Lisbeth Hybholt, Olivia A. Meuser, Jessica Eustace Cook, Carmel Downes, Jean Morrissey

**Affiliations:** 1School of Nursing and Midwifery, Trinity College Dublin, D02PN40 Dublin, Ireland; eustacj@tcd.ie (J.E.C.); cadownes@tcd.ie (C.D.); morrisje@tcd.ie (J.M.); 2Mental Health Services East, Psychiatry Region Zealand, 4200 Slagelse, Denmark; lihy@regionsjaelland.dk; 3Psychiatric Research Unit, Psychiatry Region Zealand, 4200 Slagelse, Denmark; 4College of Arts & Sciences, University of Pennsylvania, Philadelphia, PA 19102, USA; omeuser@sas.upenn.edu

**Keywords:** bereavement, suicide, peer interventions, scoping review, postvention

## Abstract

Suicide bereavement support groups are a widely available format of postvention service. Although other reviews have addressed peer-led bereavement interventions, no review has focused specifically on peer-led support for people bereaved by suicide. Informed by a framework for undertaking scoping reviews, we conducted a systematic review according to PRISMA-ScR guidelines. Searches conducted in May 2021 of peer-reviewed literature in MEDLINE (EBSCO), CINAHL Complete (EBSCO), PsycINFO (EBSCO), EMBASE (Elsevier), AMED (EBSCO), ERIC (EBSCO), Web of Science (Core Collection), ASSIA (Proquest), and Global Index Medicus. The search was not limited by language, and all studies were included to full text screening. The search identified 10 studies conducted between 1994 and 2020 in five countries. The selected papers were subjected to quality assessment. The interventions included face-to-face groups, telephone and online groups/forums and were evaluated using a variety of methodologies, which made comparison and synthesis challenging. Thematic analysis resulted in four themes: motivation, impact, aspects of intervention which hindered/enhanced outcomes, and recommendations for the practice of peer support made by the authors. While there were methodological limitations to most studies included in this review; the studies do indicate the potential benefit of peer-led support to those bereaved through suicide. Future studies should provide a definition of ‘peer’ and a clear description of the intervention being evaluated. As the field matures there is a need for more rigorous evaluation of peer interventions with representative samples, studies that compare the impact of various types of peer interventions, and studies of the peer group processes.

## 1. Introduction

Death by suicide is an extremely complex issue that impacts hundreds of thousands of people every year globally, with estimates suggesting that almost 700,000 people die from suicide every year [1,2]. A suicide death not only impacts the wellbeing of close family members and friends but also affects many people outside of this immediate circle, including neighbours, passers-by, or professionals caring for the person.

It is estimated that 135 individuals are exposed by every suicide death [3]. Cerel et al. [4] provides a nested model of suicide survivorship. The outer circle of the model encompasses all those who have been ‘exposed’, defined as anyone who knows or identifies with someone who dies by suicide and within this group there are those that are ‘affected’ by the loss. Within the affected group are two further subgroups comprising those who have an attachment relationship to the deceased and experience ‘short term’ or ‘long-term’ distress.

Coping and adapting to a loss by suicide can be particularly challenging owing to feelings of guilt, responsibility, shame, and rejection [5,6] and may have long-lasting impact on physical and mental health, including increased risk of suicidal thoughts and behaviours [7,8,9]. The bereavement may also have long-lasting consequences for families as well as individuals, altering patterns of communications within the family unit and contributing to the loss of cohesion and relationship breakdown [10]. Hence, one of the key priorities within suicide prevention policy and strategy (for example [1,11,12,13,14]) is the provision of a range of supports, both informal and formal, to help those affected by suicide and suicide behaviour to navigate the grieving process and reduce the risk of suicide and other adverse effects [15]. In addition, statement five in the National Institute of Health and Care Excellence’s Suicide Prevention Quality Standard focuses on ‘supporting people bereaved or affected by suspected suicide’ [16].

Suicide bereavement support groups are a widely available postvention service. Frequently initiated by people bereaved by suicide, they are often based on the principles of sharing experiences and offering mutual assistance, with the aim of reducing distress and risk of mental and emotional problems. Despite the increasing use of peer support groups as well as user demand for peer-delivered services, Rawlinson et al. [17], notes that there was scant attention given to peer support for suicide bereavement as an intervention, and no evidence regarding impact or effectiveness. To the authors’ knowledge, five systematic reviews of interventions for people bereaved by suicide have been published to date [5,6,18,19,20]. Four of these reviews focus on the effectiveness of individual and group-based interventions delivered in a school, family, and community context and facilitated by health care professionals, researchers, or health care professionals in conjunction with volunteers. Although the reviews point to some evidence of positive impact of the interventions on mental health and grief outcomes [5,6,18,20], few of these reviews included peer-led interventions. In two of the reviews, only controlled studies were included [5,20] while Linde et al. [6] only included studies with quantifiable outcome measures.

Three reviews were found that addressed peer-led interventions. One scoping review focused on peer support programs (excluding bereavement programs) that aim to reduce suicidality in people deemed at risk [21], and another scoping review focusing on describing the breadth of peer-delivered suicide preventions services and their outcomes, to inform future service delivery and research [22]. Bartone et al.’s [19] systematic review focused on peer support services for bereaved survivors, irrespective of the cause of death. Whilst this review provides evidence of beneficial impacts of peer interventions in terms of reduced grief symptoms, depression and suicidal thinking, as well as enhanced well-being and personal growth, and is helpful in understanding the benefits to those bereaved by suicide, it does not focus exclusively on peer support for suicide bereaved, with only seven of the thirty-two studies included involving those bereaved by suicide, including members of community mental health teams and counsellors bereaved by suicide. Of the seven studies included, the authors state that in six, support was provided by others who were also bereaved by suicide. However, it is unclear how the term peer was conceptualised and defined, or what types of peer intervention were provided. Given the distinctive nature of bereavement by suicide, in terms of stigma, self-blame, guilt, societal reactions [6], the potential for prolonged and complicated grief [23], and the absence of a review in this area, the authors were of the view there was merit in conducting a separate review to examine the extent, range, and nature of research activity in this area. Pooling data and sharing information and learnings from a review is also important for people who are involved in developing peer interventions and preparing peer facilitators.

### Aims of the Review

This scoping review was informed by Arskey and O’Malley’s [24] paper on scoping reviews and aimed to examine peer-led interventions for people bereaved by suicide. Using a systematic process following PRISMA guidelines the objectives of the review were to: (i) describe how peer is conceptualised and defined; (ii) discuss models of peer-led interventions used; (iii) describe the outcomes of peer-led interventions; (iv) identify elements of peer-led intervention that enhanced or hindered outcomes.

## 2. Methods

A systematic search of the following electronic databases was undertaken by the librarian (J E-C): MEDLINE (EBSCO), CINAHL Complete (EBSCO), PsycINFO (EBSCO), EMBASE (Elsevier), AMED (EBSCO), ERIC (EBSCO), Web of Science (Core Collection), ASSIA (Proquest), and Global Index Medicus (WHO). These included the main databases used in any health sciences-related systematic review, as well as important databases for educational and social sciences research.

Previous reviews and the authors’ knowledge were used to determine keywords, for example, terms denoting suicide (e.g., killing oneself), bereavement (e.g., loss, mourn, grief), as well as peer support (e.g., self-help, social support, peer group). In all cases, these terms were searched for, in titles and abstracts, and, where appropriate, other fields such as the “contributed indexing” field in MEDLINE (EBSCO). These were combined with controlled vocabulary terms such as MeSH, EMTREE, and CINAHL Headings as appropriate (see example MEDLINE search strategy, in Appendix A).

The search was run from the inception of the database and limited to peer-reviewed papers published before May 2021. These search boundaries resulted in 13,663 papers. A further 227 articles were located through a grey literature search, which resulted in a total of 13,890 papers. Endnote was used to screen the majority of duplicates (6987) and a further 123 duplicates were removed on import into Covidence. After duplicates were removed, the resulting 6780 papers were screened according to the following inclusion and exclusion criteria (see Table 1).

Covidence screening software (www.covidence.org (accessed on 30 June 2021)) was used to manage the screening process. Two reviewers independently assessed each title and abstract against the inclusion/exclusion criteria identified in Table 1 to identify potentially relevant papers (LH, AH, OM, JM, each pair assessed 50% or 3390 papers) and any discrepancies were resolved by a third reviewer not involved in screening that paper. For stage two screening, the full texts of 115 papers were obtained and assessed independently by two reviewers, one person assessed all (LH, AH, OM, JM). Any discrepancies at this stage were resolved by discussion with the wider team. This stage resulted in the exclusion of a further 105 papers, primarily due to their not being focused on peer-led intervention, not focused on bereavement because of suicide, not being primary research, or including those not bereaved by suicide. Following this, reference lists in these papers were reviewed, which resulted in no new additions, resulting in 10 studies in the final review. Figure 1 shows the PRISMA diagram.

## 3. Data Extraction and Analysis

A Microsoft Excel data extraction form was developed and piloted for extracting information from each study. Information on author, year, country of origin, study aim, sample achieved, population characteristics, and core data on methodology, outcomes of peer support, and factors that enhanced or hindered outcomes and findings were extracted and inputted into the Excel form (see Appendix A for overview of studies and Appendix A for more information on findings). In line with the aims of the review, information related to how peer support was defined/conceptualised, the nature and type of the peer intervention, as well as recommendations for further study in the area were also extracted. Following this, data were coded inductively into themes in line with the aims of the review. To ensure consistency, data extraction for each paper was completed independently by at least two people (LH, AH, CD, OM).

## 4. Quality Appraisal and Quality of the Studies

While quality appraisal may or may not be part of a scoping review, given the relative newness of research on peer interventions, we chose to include a quality appraisal element to help inform and improve the quality of future research. Two reviewers independently scored each paper and came together to agree the final score using the categories, weak, moderate, and strong. To support the assessment of quality, quantitative studies were reviewed using the Effective Public Health Practice Project Quality Assessment tool [25] and the Critical Appraisal Skills Programme Qualitative Critical Appraisal Tool (CASP) (https://casp-uk.net/casp-tools-checklists/ (accessed on 9 July 2021)) was completed on studies using qualitative methods (see Appendix A for more information on the tools).

Overall, the body of evidence was of mixed quality. The main reasons for the low/weak scoring in the qualitative studies were: insufficient information on issues such as ethics and data analysis, and lack of discussion on relevance/transferability of research to other contexts or discussion on reflexivity and influence of the researcher on the research, and vice-versa. In addition, the quality of the supporting data was sparse in some studies.

For the studies which were quantitative or had a quantitative component, half of them did not report receiving ethical approval or the consent of participants. Despite collection of data using a range of existing survey tools or designed measures to capture data, such as the benefits and limitations of internet forums [26], or helpfulness and comfort of meetings [27], only a few of the studies commented on the reliability of the tools used. The absence on information on demographics, withdrawals, and selection bias were also reasons for weak scores.

## 5. Findings

### 5.1. Study Characteristics

The ten studies included were conducted between 1994 and 2020. Three studies were conducted in the USA [28,29,30], two in Canada [27,31], and one in England [32], Italy [33], and Sweden [34]. Two of the studies, which were online forums, contained participants from different countries. One of them comprised mostly US-based participants with smaller numbers from Canada, the UK, and Australia [26]. The second contained both Dutch and Belgian participants [35]. The study designs included six qualitative studies, three quantitative studies, involving the use of surveys [26,29,31], and one mixed method study, which used both interviews and surveys [27]. The six qualitative studies collected data using participant observation [28,33] and interviews [33]; one study used a focus group and semi structured interviews [32]; one used telephone interviews [30]; two used online forum messages as data [34,35].

The studies that were quantitative in nature or had a quantitative element [26,27,29,31] collected data using a range of existing survey tools or designed measures to capture data, such as the benefits and limitations to internet forums [26], or helpfulness and comfort of meetings [27]. Only a few of the studies commented on the reliability of the tools used [26,27,29]. In addition to the closed questions, two studies used several open-ended questions to elicit information about the helpfulness of the meetings [27] and participants’ social networks, their motivations for attending groups, and their experiences of groups [31]. All surveys were administered during the intervention, apart from Barlow et al. [27], who administered the Hogan Grief Reaction Checklist pre- and post-intervention. Analysis was conducted using descriptive statistics [26,29,31], and also inferential statistics [27,29].

### 5.2. Participant Characteristics

All studies involved people who were bereaved by suicide, with one group including three people who had tried to take their own life [33]. In terms of participants’ relationship to the deceased, two studies focused solely on parents [29,34]; two studies listed the relationships as including parents, children, spouses/partners, siblings and others, without specifying the actual number of each [27,28]; one study had parents in the majority of the sample [32], while the remaining two, which stipulated the relationship, had similar numbers of parents and partners [33,35]. In addition to parents and partners/spouses, all samples included at least some siblings and children, while other relatives and non-relatives, including friends, comprised a minority of the samples. In terms of the duration of time since suicide, the range was 6 weeks to 20 years [27,32]. One study gave the mean duration as being 3 years [32]. In most of the studies, the majority of participants had sustained the loss within the past five years [26,27,35]; only in Feigelman & Feigelman [30] were the majority of participants bereaved for more than five years.

It is not possible to report the collective sample number, as Hopmeyer & Werk [31] did not report sample size and or demographics, while the sample size in another study was the number of online messages analysed, rather than the number of participants [34]. In those that reported age, participant age ranged from 17 to 81 years [26,27,32,33]. Two studies gave the mean ages of the sample as 46.9 and 52.3 [26,27]. Age was not reported or there was incomplete reporting of age in several studies [28,29,30,31,34,35]. Over 70% of the reported samples were female [27,32,33,35], with as many as 94.6% females in one study [26]. None of the studies reported on the ethnicity, sexual orientation, or education level of participants.

### 5.3. How Peer and Peer Support Is Conceptualised and Defined

No study specifically defined what they meant by the term peer. Of the studies that briefly alluded to the concepts of ‘peer support’ or ‘peer help’ in the introductions to the papers, it is evident that peer support or peer help is regarded as an exchange of support among people who share similar experiences [27,28,32]. Feigelman & Feigelman [28], in their introduction, briefly discuss the perceived benefits and drawbacks of peer-led versus professional-led support groups, without defining either. One study described the groups being researched as “mutual aid” groups [33], while another described them as “self-help” groups [31].

### 5.4. Type of Peer Interventions Evaluated

The interventions included five face-to-face groups [29,30,31,32,33] and four online groups/forums [26,29,34,35]. One of the studies was a face-to-face or telephone peer support program in which a peer supporter was matched with a ‘client’ who had experienced the same type of loss [27]. Six studies indicated that the groups/pairings were facilitated/moderated by a peer bereaved by suicide [27,28,30,31,33,34]. The background of the person/s was not stipulated in three studies [29,32,35] while it was unclear if one of the studies involved any facilitation/moderation [26].

Three studies indicated that the groups met monthly for approximately two hours [29,30,32] while one group met weekly for 1.5 h [33] and another held bi-monthly meetings [31]. The 24/7 availability of the online groups/forums was mentioned in two studies [26,29] but presumably all four were always accessible for an indefinite period of time. Only three studies indicated the duration of the group/pairing; in the case of Barlow et al. [27], the pairing met for a period of four months, while two studies indicated that the groups were open-ended (meaning members could join at any time) [28,31].

Few studies make any reference to the format and structure of the groups/forums. One of the groups is described as “open discussion” [31]. Another described how the group opened with a brief period of informal social interaction before the peer-facilitator initiated introductions, encouraged people to contribute in a non-judgemental fashion, and ensured new members had the opportunity to speak if they so wished. The group then closed with a serenity prayer and refreshments [28,30]. One of the internet forum studies described how members were given some instructional guidance about items of interest and how best to contribute to the forum, but otherwise were free to “dialogue” with other members [29].

Whether peer facilitators/moderators were offered education and support for the role was not or poorly described in most studies. One study stipulated that the peer was trained and partook in monthly debriefing and educational sessions [27], while two indicated that the peer facilitator had the support of professionals, one of whom was a social worker, while the other study didn’t specify except to say that it was a ‘team of professionals’ [31,35].

### 5.5. Outcomes of the Intervention

Four themes were identified: motivation, impact, aspects of intervention which hindered/enhanced outcomes, and recommendations for the practice of peer support made by the authors.

#### 5.5.1. Motivation

In the five studies that reported motivation for accessing peer support, the main reasons included wanting to meet others with the same experiences, to access information and support to understanding, and coping with the suicide loss. Feelings of depression and loneliness as well as stigma from social networks prompted people to seek out a safe, understanding, and non-judgemental environment with peers. Helping and giving support to others was also a motivating factor [27,31,32,34,35]. In one study, accessing online support was linked to being ‘turned off’ by face-to-face groups or none being available [29].

#### 5.5.2. Impact

The shared understanding, empathy, and the information received from peer support impacted participants positively in terms of a reduction in self-blame, isolation, and stigma, as well as gains in hope, self-worth (by helping others), personal well-being, and a sense of connectedness. Participants reported better coping strategies, problem-solving skills, and empowerment to grieve and change. In relation to the grief process, the peer group helped with acceptance and processing the grief by enabling participants to gain a better understanding of their own and others’ experiences, normalise the loss, as well as offering the opportunity to memorialise the deceased, which was important to the bereavement process [26,27,28,29,30,31,32,33]. In addition, Silvén Hagström [34], reported that peers helped participants construct new meanings or narratives around the suicide. Only one study conducted pre- and post-testing of the intervention and reported that some grief reactions (despair, detachment, and disorganisation) were reduced at post-intervention follow-up while peer support was rated highly in terms of comfort and helpfulness [27]. Schotanus-Dijktra et al. [35] found more positive comments about the online experience when compared to negative comments (9 vs. 1%), however, most of their findings related to the nature of online interactions rather than their impact. In the few studies that reported negative impacts, these related to feelings of distress and depression as a result of being involved in the peer intervention [26,28,29,30].

### 5.6. Aspects of Intervention Which Enhanced Outcomes

Helpful aspects of interventions fell into three categories: group process, group format, and group composition. The group process was enhanced by good facilitation and leadership skills. These included facilitators being flexible and accommodating regarding time and place of meeting, ‘being present’, providing additional support, structure, and managing group diversity and dynamics effectively [27,29,30]. In addition, debriefing opportunities and support for peer leaders provided reassurance and confidence in their role [27]. In terms of format, peers bereaved by suicide appreciated the 24/7 availability, anonymity, and ‘democratic participatory style’ of internet support groups [26,29], with some participants in one study identifying it as preferable to face-to-face support [29]. While members with similar kinship relationships facilitated a connection and shared understanding to form group diversity, [27], in terms of having a mix of individuals with different kinship relationships to the deceased, including individuals at different stages of grief and individuals who had tried to take their own life, this was identified as beneficial for generating a dialogue in which different perspectives challenged ways of thinking [28,33]. It also meant that some peers who were longer bereaved could adopt a more supportive role in which they offered guidance and hope to others [32].

### 5.7. Aspects of Intervention Which Hindered Outcomes

Unhelpful aspects of interventions fell into three categories: group process, group format/structure, and logistical challenges. In terms of the group process, poor facilitation made the intervention less acceptable to participants, in some cases, causing individuals to leave or change groups. Poor facilitation included insufficient monitoring of online posts, not addressing participants’ upset, allowing some members to monopolise the group or form cliques, and spending too much emphasis on some topics and not enough on others [26,28,29,30]. In addition, two studies identified issues in relation to the management of expectations and boundaries around interactions, including how to terminate relationships and respond to people who took political and social action agendas in relation to suicide into the group [27,30]. Group format/structure issues, which affected the acceptability and effectiveness of interventions included groups being too large, lacking members with the same kinship relationship or experiences, the degree of spirituality/religiosity of the group, and group sharing/disclosure which some members found uncomfortable or upsetting [28,29,30,32]. Finally, logistical issues such as distance to travel to groups, timing, and available space were identified as challenging in two studies [27,32].

### 5.8. Recommendations for the Practice of Peer Support Made by the Authors of the Studies Included

Training and support for facilitators was regarded as essential by some authors to enable people to facilitate groups with confidence and address problems which arise in groups, such as monopolisation by some individuals, feelings of distress, how to manage different expectations and agendas of members, and how to set clear guidelines from the outset [27,28,30]. As well as developing group facilitation and leadership skills, knowledge of the bereavement process and referral resources were deemed important parts of the ongoing education of facilitators [30]. One paper recommended that support group members also be given information and tools to enable them to participate effectively, equitably, and non-judgementally in groups [28]. In terms of online forums, Bailey et al. [26] recommended the development of guidelines “governing the conceptualisation, development, and maintenance of Internet forums in order to ensure their safety and clinical utility” (p. 399) while consultation with users was advised to ensure that the forums meet the needs of different groups. Finally, Feigelman & Feigelman [30] recommended that support groups be open-ended; although they acknowledged that there may be conflicts in terms of the needs of newly bereaved people in comparison to longer term survivors, they believed that there was benefit to having both in the mix.

## 6. Discussion

The involvement of peers and peer intervention are now an acknowledged cornerstone of all mental health policy [36,37,38,39] and this review set out to examine peer-led interventions for people bereaved by suicide. Although the aim of the studies included was to evaluate a peer-led intervention, in the majority of cases, peer support was not explicitly defined, hence the authors inferred that it was understood as a form of reciprocal support between individuals who share similar experiences and as being distinct from professional support. While the type of peer-led intervention offered was typically a group format in which there was a moderator/facilitator present, the nature of the peer interventions was poorly described in terms of duration, specification of the intervention (i.e., short-term vs. long-term; open vs. closed membership), or if the interventions followed a certain structure, used ground rules, or addressed certain topics. It was also unclear if training and support was provided to the facilitators/moderators or whether people attending had to disclose their loss or identify how the person died to meet the criteria to join the group. These omissions made comparison and interpretation of findings a challenge, as similarities and differences between groups were not explicit.

Nevertheless, a recurring theme within the data was the positive impact of being able to share narratives of loss in a non-judgemental and supportive environment, wherein people, by virtue of their similar experiences, could help and support each other to process their grief. This support in turn buoyed individuals to the extent that they felt less alone and stigmatised, and more hopeful and empowered. Recent qualitative studies of the support needs of people bereaved by suicide highlight how the feelings of stigma make social interactions difficult, uncomfortable, and painful, which in turn leads to withdrawal and self-isolation [40,41]. While there were differences of opinions in terms of who should lead peer groups, participants in both of these studies iterated the importance of finding a place to talk, without fearing people’s reactions and judgement. While talking to peers and sharing experiences within a group may decrease feelings of being alone and is an important element of meaning reconstruction following a loss [23], it can also be burdensome or even retraumatisating for some people. Indeed, findings in the current review indicate that some members found group sharing and disclosure uncomfortable or upsetting, a view that is also supported by a recent study of individuals with mental health problems who reported that listening to others’ recovery narratives can be burdening, saddening, and make some people feel inadequate and disconnected [42].

Another important issue highlighted in this review relates to the preparation of peer facilitators. Although the findings suggest that a person’s experience of a peer support group is highly influenced by the facilitation of the group, the nature and type of preparation provided to the facilitators was not made explicit. This may be the reason why many of the authors recommended that facilitators be provided with education to help them manage issues related to group dynamics, as well as the distress that some participants will inevitably experience. The preparation and on-going support for facilitators is also in line with many of the guidelines on bereavement support that have been published internationally (see [43,44,45]). While the authors of the studies included in the review did not question whether people bereaved by suicide should ever facilitate groups, bereaved participants in Ross et al.’s [41] recent study were of the view that people bereaved by suicide may not necessarily be able to lead a group. Hence, they were of the view that trained professionals should facilitate the support groups to ensure a safe and helpful dynamic in the group.

In terms of online support, although a small number of people commented on inadequate moderation and the posting of inaccurate information, online peer support interventions emerged as having advantages over face-to face peer interventions in terms of the accessibility and anonymity offered. Thus, the reported acceptability and benefits from using online support forums highlight this as a type of peer support with potential, however, similar to face-to-face groups, issues around quality and safety need to be addressed. Only one of the studies included in the review compared online peer group support with face-to-face peer group support [29], and although they found that grief symptoms were higher in the online community, they were unable to attribute this to any deficiency in online support but rather identified greater stigmatisation from family and friends as a possible reason for the discrepancy.

In addition to education and preparation, the review also identifies several gaps that need consideration in further research. To enhance future comparison and synthesis of research findings, there is a need for researchers to clearly define what they mean by peer and describe in detail the intervention being evaluated. Without methodologically rigorous studies involving control groups using valid outcomes measures, that not only relate to grief and mental health outcomes, but also outcomes such as hope, empowerment, stigma, coping and impact on life outside the group etc., it is difficult to determine the true impact of peer-led support interventions. The comparative impact of various types of peer interventions (face to face/online support) versus other types of support, such as those offered by professionals, also warrants study. Existing studies do not shed light on what is the optimal length of time since the loss before one may benefit from a peer support intervention, or indeed what the optimal length of time is for support.

Further research is also needed in the form of observational studies that explore group processes in real time, for example, how groups practice and maintain helpful elements such as a non-judgemental approach, generate feelings of connectedness, and how they share stories and thoughts to help and support each other. The potential for distress from being exposed to individuals’ narratives of loss also warrants attention, as well as the reasons people leave peer groups. Additionally, the impact of different biographical backgrounds (relationships to the deceased, age, gender, and different stages of grief) on group processes and outcomes also warrants further attention.

There is also a need for research with more representative samples to determine if results can be generalised. The lack of representation of men and older people (those aged over 65) in the samples begs the question of whether the interventions would have demonstrated the same utility for these cohorts, particularly as men are believed to employ different coping strategies in response to bereavement in comparison to women [46]. Indeed, Hopmeyer & Werk’s [31] study, albeit dated, indicates that men may be more focused on problem solving than sharing emotions within peer support groups. It is also not known if the findings could be replicated across cultural contexts, given that many of the peer interventions were based in North America and Western Europe. The absence of longitudinal research in this area is a research gap, which also needs to be filled in order to identify causal factors affecting grief experiences overtime. Only one of the studies explored factors contributing to departures from groups [30]; the authors in this paper recognise this as an area which requires further attention. In terms of online forums, the quality of the forums and the impact of this on efficacy, the perceptions of different groups (users, moderators, professionals) about their efficacy, and the potential for distress are also areas for further research.

Although the review has several strengths such as its breadth of questions, the use of a comprehensive multi-database search strategy, and dual-author data extraction and analysis, it needs to be read with the following limitations in mind. First, some relevant papers may have been missed because of the exclusion of non-English literature. Second, as discussed, most of the studies included in the review were of low or moderate quality. Third, the potential for interpretative bias impacting the findings is also an issue, as many papers did not clearly define the meaning of terms, such as peer, or provide detailed information on the intervention being evaluated.

## 7. Conclusions

While there were clear methodological limitations to many of the studies included in this review, the studies do indicate the potential benefit of peer-led support to those bereaved through suicide. The review revealed an unclear conceptualising of peers and the peer facilitators of the support group. It became clear that the role of the facilitator was very important and could influence the group process. However, it was not clear in each study how the peer facilitators were prepared for the role and how this may influence the outcome. This field of study would benefit from more in-depth description of the interventions provided, including how facilitators are selected and prepared to facilitate groups. In addition, larger studies with more representative samples to allow for comparisons across groups, including professionally facilitated peer groups, is required. Research to explore the long-term impact of peer-led intervention, including people’s reasons for leaving peer support groups is also needed.

The findings also suggest that not all participants may benefit from partaking in a group, as some experienced the process of sharing upsetting. This suggests a need for further investigation into this issue, including the impact of facilitator style and training on this outcome. In line with this, it is important to explore how helpful processes arise and are maintained in a peer-led group and how attending a peer-led group influences everyday life, such as feelings of stigma and recovery in the aftermath, as support groups are supposed to enhance life outside the group.

## Figures and Tables

**Figure 1 ijerph-19-03485-f001:**
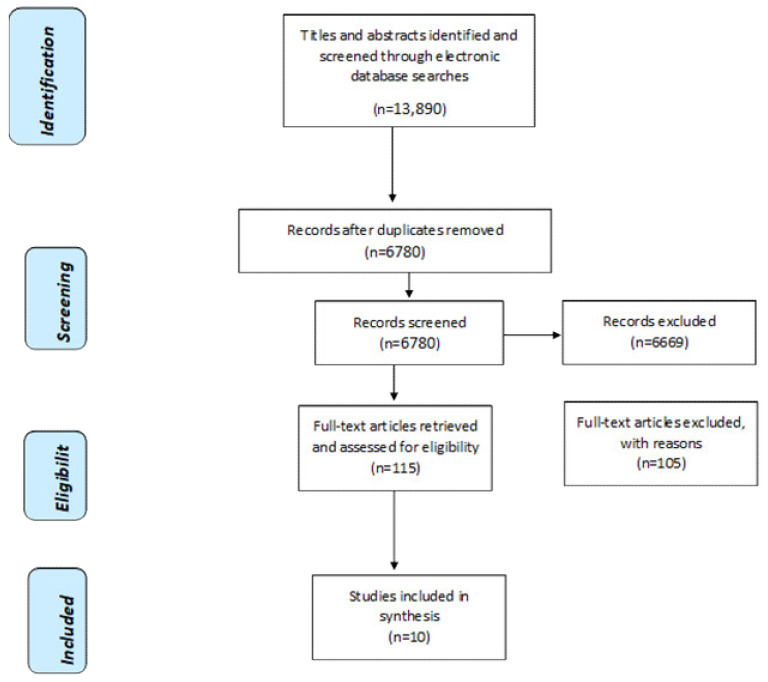
PRISMA Flow Diagram of Selection Process.

**Table 1 ijerph-19-03485-t001:** Inclusion/exclusion criteria.

Inclusion
i. Empirical studies using any research design
ii. Studies focusing on peer-led interventions
iii. Target population -people bereaved by suicide
iv. Conducted in any country, location, and setting, and using any modes of delivery (online, face to face, phone etc.)
v. Reported on any outcomes (e.g., feasibility, acceptability, effectiveness etc.)
Exclusion
i. Descriptive or theoretical papers focused on a peer intervention without evaluation findings
ii. Studies focused on evaluating peer interventions for bereavement (including bereavement by suicide) where it was not possible to extract information specific to bereavement by suicide
iii. Studies of interventions for bereavement (including bereavement by suicide) where there was a co-facilitation element by a professional
iv. Literature reviews, systematic reviews, discussion papers, opinion articles/editorials, commentaries, book chapters, conference papers, and case studies (*n* = 1).

## Data Availability

Not applicable.

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
