# Peer review of "Scoping Review of Peer-Led Support for People Bereaved by Suicide"

_ijerph, 2022, doi:10.3390/ijerph19063485_

Round 1

Reviewer 1 Report

This manuscript presents a scoping review of evaluations of peer-led support for people bereaved by suicide. This is a timely and pertinent question to examine the impacts of peer support and examine what learning can be extracted from existing literature in order to improve these services for this vulnerable group. It is clear that the authors have done a thorough job of reviewing the literature; I do, however, feel that the messages of the paper could be stronger with some re-structuring, and more focus on the real-world impact of their conclusions.

The current formatting of the manuscript is weighted towards both methodology (i.e. how the authors performed their scoping review), and quality assurance of the included papers. I think because of this the important messages derived from the review get lost (currently page 16/22 before any substantive discussion of the outcomes of the interventions). I have some suggestions for restructuring that I believe would improve the ‘narrative’ of the work from justification of the paper and it’s place in the literature to conclusions that may be relevant to improve these groups.

Introduction – I appreciate that the authors have noted the particular reasons why examining peer-support for people bereaved by suicide (as opposed to any other type of bereavement) is needed, i.e. that this is a potentially different and complex grief. This work is timely due to the increased emphasis in policy (I’m thinking particularly of England as this is my area of knowledge) of support for people bereaved by suicide being a key goal of the national suicide prevention plan, and a key standard in the NICE quality standard for suicide prevention. The Office for Health Improvements and Disparities (OHID – formerly Public Health England) is working towards a national real-time surveillance of suicide system, which will facilitate the provision of bereavement support in local areas, as set out in the NHS Long-Term Plan. This review work would benefit from being set in the context of these current developments in suicide bereavement support – there will be many new groups established, and ideally they would learn from the literature as to what may or may not be effective. It’s worth also noting in the introduction that much of the push for bereavement support for people bereaved by suicide has been led by the tireless campaigning of people bereaved by suicide – this implies a motivation in many people bereaved by suicide for these types of support groups.

Methodology – I would prefer to see a briefer methodology section, though I appreciate how detailed your work has been. Could table 1 be a supplementary table, perhaps with key search terms cited in the text itself?

Quality appraisal – I am not convinced that the quality appraisal adds to your paper, these tools to my knowledge are designed to assess studies that were conceived as scientific studies, and don’t fit with support groups. I think the authors agree as stated in the abstract “The interventions included face-to-face groups, telephone and online groups/forums and were evaluated using a variety of methodologies, which made comparison and synthesis challenging”. Should you wish to keep the quality appraisal in your paper, I suggest this could be another aim – something like 'Can peer-led support groups for people bereaved by suicide be effectively evaluated?' And then look at the methods used for this purpose.

Table 3 - I would greatly reduce the information included in table 3, and – if you wish – include the full table as a supplementary table. As it is, this table spans 6 pages and has no headings following through from the first to subsequent pages. I don’t think the reader needs to know all of the information included here, and – as with the amount of detail on the methodology – this amount of detail detracts from what I believe to be the key messages of the paper from page 16 (outcomes and recommendations).

Findings – I would reduce the word count of the study and patient characteristics. In the current manuscript the reader has reached page 14 before you have begun discussion of your first aim (to define peer support).

Outcomes – Section 9 (page 16) to me is the centre of your paper. I would like to see this as the central element of your findings, quality appraisal reduced (or taken out altogether), sensitive context given (including acknowledgement of lived experience in driving support for people bereaved by suicide at local and policy-level), and then real-world implications of the findings from this section discussed.

Discussion – Ideally this would begin with an overview of what you consider to be the key messages extracted from your scoping review (for me this would be a brief summary of the outcomes) rather than the aims and then straight into the limitations of your work.

Conclusions – I appreciate that this is a scoping review, however I feel that your conclusions don’t necessarily need to be limited to determining future research focus. Could you make some suggestions as to what implications there are from your findings? i.e. the facilitator role is very important and needs exploring further – how does this fit with existing guidelines for delivering bereavement support groups? You mention the need to explore negative impacts of peer-led suicide bereavement support – how does this fit with guidelines on timeliness of bereavement counselling for example? Particularly in relation to the set-up of suicide bereavement support being offered as part of real-time surveillance systems (i.e. almost immediately available).

I hope you find these suggestions helpful to strengthen the narrative through your paper.

Author Response

We would like to thank the reviewers for their very helpful comments. We have tracked changes with our additions. However, in the interest of ease of reading we didn’t use track changes for what we deleted, we just deleted the information.

Reviewer 1

1 Comment: This manuscript presents a scoping review of evaluations of peer-led support for people bereaved by suicide. This is a timely and pertinent question to examine the impacts of peer support and examine what learning can be extracted from existing literature in order to improve these services for this vulnerable group. It is clear that the authors have done a thorough job of reviewing the literature. I do, however, feel that the messages of the paper could be stronger with some re-structuring, and more focus on the real-world impact of their conclusions. The current formatting of the manuscript is weighted towards both methodology (i.e., how the authors performed their scoping review), and quality assurance of the included papers. I think because of this the important messages derived from the review get lost (currently page 16/22 before any substantive discussion of the outcomes of the interventions). I have some suggestions for restructuring that I believe would improve the ‘narrative’ of the work from justification of the paper and its place in the literature to conclusions that may be relevant to improve these groups.

Response Thank you for this positive comment and for you very helpful and supportive suggestions

2 Comment:  Introduction – I appreciate that the authors have noted the particular reasons why examining peer-support for people bereaved by suicide (as opposed to any other type of bereavement) is needed, i.e., that this is a potentially different and complex grief. This work is timely due to the increased emphasis in policy (I’m thinking particularly of England as this is my area of knowledge) of support for people bereaved by suicide being a key goal of the national suicide prevention plan, and a key standard in the NICE quality standard for suicide prevention. The Office for Health Improvements and Disparities (OHID – formerly Public Health England) is working towards a national real-time surveillance of suicide system, which will facilitate the provision of bereavement support in local areas, as set out in the NHS Long-Term Plan. This review work would benefit from being set in the context of these current developments in suicide bereavement support –

Response we have added the following to the introduction

Hence, one of the key priorities within suicide prevention policy and strategy (for example the World Health Organization, 2102, 2014, 2021; Department of Health (UK), 2012; Department of Health (Irl), 2015) is the provision of a range of supports, both informal and formal, to help those affected by suicide and suicide behavior to navigate the grieving process and reduce the risk of suicide and other adverse effects (Aguirre and Slater, 2010). In addition, statement five in the National Institute of Health and Care Excellence’s Suicide Prevention Quality Standard focuses on ‘supporting people bereaved or affected by suspected suicide’ (NICE,2019).

3 Comment:  There will be many new groups established, and ideally, they would learn from the literature as to what may or may not be effective. It’s worth also noting in the introduction that much of the push for bereavement support for people bereaved by suicide has been led by the tireless campaigning of people bereaved by suicide – this implies a motivation in many people bereaved by suicide for these types of support groups.

Response: we have added the following to the rationale in the introduction

Pooling data and sharing information and learnings from a review is also important for people who are involved in developing peer interventions and preparing peer facilitators.

4 Comment: Methodology – I would prefer to see a briefer methodology section, though I appreciate how detailed your work has been. Could table 1 be a supplementary table, perhaps with key search terms cited in the text itself?

Response – we have amended the paper, and this is now a supplemental table 1

5 Comment: Quality appraisal – I am not convinced that the quality appraisal adds to your paper, these tools to my knowledge are designed to assess studies that were conceived as scientific studies, and don’t fit with support groups. I think the authors agree as stated in the abstract “The interventions included face-to-face groups, telephone and online groups/forums and were evaluated using a variety of methodologies, which made comparison and synthesis challenging”. Should you wish to keep the quality appraisal in your paper, I suggest this could be another aim – something like 'Can peer-led support groups for people bereaved by suicide be effectively evaluated?' And then look at the methods used for this purpose.

Response – We have  also shortened the section on quality and made it explicit that the reason we are doing this is to help inform and improve the quality of future research.

6 Comment Table 3 - I would greatly reduce the information included in table 3, and – if you wish – include the full table as a supplementary table. As it is, this table spans 6 pages and has no headings following through from the first to subsequent pages. I don’t think the reader needs to know all of the information included here, and – as with the amount of detail on the methodology – this amount of detail detracts from what I believe to be the key messages of the paper from page 16 (outcomes and recommendations).

Response – We have moved this to supplementary table 2

7 Comment. Findings – I would reduce the word count of the study and patient characteristics. In the current manuscript the reader has reached page 14 before you have begun discussion of your first aim (to define peer support).

Response – we have significantly reduced the information in this section and include it in supplementary table 2

8 Comment. Outcomes – Section 9 (page 16) to me is the centre of your paper. I would like to see this as the central element of your findings, quality appraisal reduced (or taken out altogether), sensitive context given (including acknowledgement of lived experience in driving support for people bereaved by suicide at local and policy-level), and then real-world implications of the findings from this section discussed.

Response –   as we have reduced the other sections this is now the core of the paper.

9 Comment Discussion – Ideally this would begin with an overview of what you consider to be the key messages extracted from your scoping review (for me this would be a brief summary of the outcomes) rather than the aims and then straight into the limitations of your work.

Response –   We have rewritten parts of the discussion to address above comment and moved limitation to the end of the discussion

10 Comment: Conclusions – I appreciate that this is a scoping review, however I feel that your conclusions don’t necessarily need to be limited to determining future research focus. Could you make some suggestions as to what implications there are from your findings? i.e., the facilitator role is very important and needs exploring further – how does this fit with existing guidelines for delivering bereavement support groups? You mention the need to explore negative impacts of peer-led suicide bereavement support – how does this fit with guidelines on timeliness of bereavement counselling for example? Particularly in relation to the set-up of suicide bereavement support being offered as part of real-time surveillance systems (i.e., almost immediately available).

Response –   Thank you for this comment, in an attempt to address this, we have included more comments on education and training and guidelines in this area. However, as the studies did not provide information on this aspect, we need to be cautious not to overstep the findings of the review.

We would like to thank the reviewers for their very helpful comments. We have tracked changes with our additions. However, in the interest of ease of reading we didn’t use track changes for what we deleted, we just deleted the information.

Reviewer 1

1 Comment: This manuscript presents a scoping review of evaluations of peer-led support for people bereaved by suicide. This is a timely and pertinent question to examine the impacts of peer support and examine what learning can be extracted from existing literature in order to improve these services for this vulnerable group. It is clear that the authors have done a thorough job of reviewing the literature. I do, however, feel that the messages of the paper could be stronger with some re-structuring, and more focus on the real-world impact of their conclusions. The current formatting of the manuscript is weighted towards both methodology (i.e., how the authors performed their scoping review), and quality assurance of the included papers. I think because of this the important messages derived from the review get lost (currently page 16/22 before any substantive discussion of the outcomes of the interventions). I have some suggestions for restructuring that I believe would improve the ‘narrative’ of the work from justification of the paper and its place in the literature to conclusions that may be relevant to improve these groups.

Response Thank you for this positive comment and for you very helpful and supportive suggestions

2 Comment:  Introduction – I appreciate that the authors have noted the particular reasons why examining peer-support for people bereaved by suicide (as opposed to any other type of bereavement) is needed, i.e., that this is a potentially different and complex grief. This work is timely due to the increased emphasis in policy (I’m thinking particularly of England as this is my area of knowledge) of support for people bereaved by suicide being a key goal of the national suicide prevention plan, and a key standard in the NICE quality standard for suicide prevention. The Office for Health Improvements and Disparities (OHID – formerly Public Health England) is working towards a national real-time surveillance of suicide system, which will facilitate the provision of bereavement support in local areas, as set out in the NHS Long-Term Plan. This review work would benefit from being set in the context of these current developments in suicide bereavement support –

Response we have added the following to the introduction

Hence, one of the key priorities within suicide prevention policy and strategy (for example the World Health Organization, 2102, 2014, 2021; Department of Health (UK), 2012; Department of Health (Irl), 2015) is the provision of a range of supports, both informal and formal, to help those affected by suicide and suicide behavior to navigate the grieving process and reduce the risk of suicide and other adverse effects (Aguirre and Slater, 2010). In addition, statement five in the National Institute of Health and Care Excellence’s Suicide Prevention Quality Standard focuses on ‘supporting people bereaved or affected by suspected suicide’ (NICE,2019).

3 Comment:  There will be many new groups established, and ideally, they would learn from the literature as to what may or may not be effective. It’s worth also noting in the introduction that much of the push for bereavement support for people bereaved by suicide has been led by the tireless campaigning of people bereaved by suicide – this implies a motivation in many people bereaved by suicide for these types of support groups.

Response: we have added the following to the rationale in the introduction

Pooling data and sharing information and learnings from a review is also important for people who are involved in developing peer interventions and preparing peer facilitators.

4 Comment: Methodology – I would prefer to see a briefer methodology section, though I appreciate how detailed your work has been. Could table 1 be a supplementary table, perhaps with key search terms cited in the text itself?

Response – we have amended the paper, and this is now a supplemental table 1

5 Comment: Quality appraisal – I am not convinced that the quality appraisal adds to your paper, these tools to my knowledge are designed to assess studies that were conceived as scientific studies, and don’t fit with support groups. I think the authors agree as stated in the abstract “The interventions included face-to-face groups, telephone and online groups/forums and were evaluated using a variety of methodologies, which made comparison and synthesis challenging”. Should you wish to keep the quality appraisal in your paper, I suggest this could be another aim – something like 'Can peer-led support groups for people bereaved by suicide be effectively evaluated?' And then look at the methods used for this purpose.

Response – We have  also shortened the section on quality and made it explicit that the reason we are doing this is to help inform and improve the quality of future research.

6 Comment Table 3 - I would greatly reduce the information included in table 3, and – if you wish – include the full table as a supplementary table. As it is, this table spans 6 pages and has no headings following through from the first to subsequent pages. I don’t think the reader needs to know all of the information included here, and – as with the amount of detail on the methodology – this amount of detail detracts from what I believe to be the key messages of the paper from page 16 (outcomes and recommendations).

Response – We have moved this to supplementary table 2

7 Comment. Findings – I would reduce the word count of the study and patient characteristics. In the current manuscript the reader has reached page 14 before you have begun discussion of your first aim (to define peer support).

Response – we have significantly reduced the information in this section and include it in supplementary table 2

8 Comment. Outcomes – Section 9 (page 16) to me is the centre of your paper. I would like to see this as the central element of your findings, quality appraisal reduced (or taken out altogether), sensitive context given (including acknowledgement of lived experience in driving support for people bereaved by suicide at local and policy-level), and then real-world implications of the findings from this section discussed.

Response –   as we have reduced the other sections this is now the core of the paper.

9 Comment Discussion – Ideally this would begin with an overview of what you consider to be the key messages extracted from your scoping review (for me this would be a brief summary of the outcomes) rather than the aims and then straight into the limitations of your work.

Response –   We have rewritten parts of the discussion to address above comment and moved limitation to the end of the discussion

10 Comment: Conclusions – I appreciate that this is a scoping review, however I feel that your conclusions don’t necessarily need to be limited to determining future research focus. Could you make some suggestions as to what implications there are from your findings? i.e., the facilitator role is very important and needs exploring further – how does this fit with existing guidelines for delivering bereavement support groups? You mention the need to explore negative impacts of peer-led suicide bereavement support – how does this fit with guidelines on timeliness of bereavement counselling for example? Particularly in relation to the set-up of suicide bereavement support being offered as part of real-time surveillance systems (i.e., almost immediately available).

Response –   Thank you for this comment, in an attempt to address this, we have included more comments on education and training and guidelines in this area. However, as the studies did not provide information on this aspect, we need to be cautious not to overstep the findings of the review.

Reviewer 2 Report

I think it's a meaningful paper at a time when concerns about suicide are growing. It is not clearly understood how the literature was extracted and on what reason the literature was excluded and included. Also, it is described as low/weak, moderate/medium, and high/strong, so I am curious about the  score for the quality assessment. The study objectives, results, and discussions were described relatively clearly.But I don't know if the use of the term ‘model’ is appropriate. 

Author Response

We would like to thank the reviewers for their very helpful comments. We have tracked changes with our additions. However, in the interest of ease of reading we didn’t use track changes for what we deleted, we just deleted the information.

Reviewer 2

  1. Comment: I think it's a meaningful paper at a time when concerns about suicide are growing.

Response: Thank you for this comment

2 Comment: It is not clearly understood how the literature was extracted and on what reason the literature was excluded and included.

Response – we have reworded this section to clearly include a reference to the table where the reasons for inclusion or exclusion are identified. The sentence now reads as ‘Two reviewers independently assessed each title and abstract against the inclusion/exclusion criteria identified in table 1 to identify potentially relevant papers (LH, AH, OM, JM, each pair assessed 50% or 3,390 papers) and any discrepancies were resolved by a third reviewer not involved in screening that paper’.

3 Comment Also, it is described as low/weak, moderate/medium, and high/strong, so I am curious about the score for the quality assessment.

Response – The following information on scoring is included in the supplementary table 2

The Critical Appraisal Skills Programme Qualitative Critical Appraisal Tool (CASP) (https://casp-uk.net/casp-tools-checklists/) consists of ten items (clarity of aims, appropriateness of design methodology and recruitment, rigor of data collection and analysis, ethical issues, relationship between researcher and participants, clarity, adequacy, and relevance of findings) which are scored on a scale of 0-1 and summed to determine a total CASP score. The maximum score possible score of 10, with scores 0-4.5 points considered low/weak, score of 5-7.5 points were medium/moderate quality and a scores of 8-10 points was considered high/strong quality. The Effective Public Health Practice Project Quality Assessment tool (Thomas et al 2004) was used for studies using quantitative design. This tool also had 10 domains (selection bias, type of design, confounders, blinding, withdrawals, validity and reliability of data collection, sampling, type of data, validity of evaluation instrument, intervention integrity, appropriateness of data analysis). Each domain was rated using the categories of strong, moderate, or weak. Those awarded a strong rating have no weak ratings, with those given a moderate or weak rating having one weak rating or two or more WEAK ratings, respectively.

4 Comment: The study objectives, results, and discussions were described relatively clearly. But I don't know if the use of the term ‘model’ is appropriate. 

Response – We have changed the word ‘model’ to type throughout 

Reviewer 3 Report

Thank you for the invitation to review this excellent scoping review.  I have a few brief comments.

The introduction and methods were thorough, however, it was stated that two reviewers independently assessed each title and abstract against inclusion/exclusion criteria yet there were initials of four authors. I am assuming that this means no one author reviewed all 6780 titles and abstracts and perhaps each of the four reviewed half of them? Thank you for clarifying.

There seemed to be information lacking about what constitutes suicide among the bereaved.  Did bereaved participants have to clearly identify their loss in order to meet criteria to join a group?  What happened if there was a discrepancy regarding the death, for example, if it was unclear if the person who died did not intentionally kill themselves? 

It is also unclear if there were complications/challenges amongst bereaved participants if there were large age ranges or differences within the groups. An example of this might be with family members of a person who died by suicide due to suffering from a chronic or serious illness, who may be in a group with bereaved parents or siblings of a younger person suffering from depression or other mental illness. Could this be added as a limitation or concern that might be addressed?

Overall, I found this review to be extremely well written and recommend it for publication.

Author Response

We would like to thank the reviewers for their very helpful comments. We have tracked changes with our additions. However, in the interest of ease of reading we didn’t use track changes for what we deleted, we just deleted the information.

Reviewer 3

1 Comment:  Thank you for the invitation to review this excellent scoping review.  Overall, I found this review to be extremely well written and recommend it for publication.

Response: Thank you for this comment

2 Comment: The introduction and methods were thorough; however, it was stated that two reviewers independently assessed each title and abstract against inclusion/exclusion criteria yet there were initials of four authors. I am assuming that this means no one author reviewed all 6780 titles and abstracts and perhaps each of the four reviewed half of them? Thank you for clarifying.

Response: we have reworded this section. The sentence now reads as ‘Two reviewers independently assessed each title and abstract against the inclusion/exclusion criteria identified in table 1 to identify potentially relevant papers (LH, AH, OM, JM, each pair assessed 50% or 3,390 papers) and any discrepancies were resolved by a third reviewer not involved in screening that paper.’

3 Comment: There seemed to be information lacking about what constitutes suicide among the bereaved.  Did bereaved participants have to clearly identify their loss in order to meet criteria to join a group?  What happened if there was a discrepancy regarding the death, for example, if it was unclear if the person who died did not intentionally kill themselves? 

Response: Unfortunately, the papers do not provide this information and we now have included a reference to this in the discussion

4 Comment: It is also unclear if there were complications/challenges amongst bereaved participants if there were large age ranges or differences within the groups. An example of this might be with family members of a person who died by suicide due to suffering from a chronic or serious illness, who may be in a group with bereaved parents or siblings of a younger person suffering from depression or other mental illness. Could this be added as a limitation or concern that might be addressed?

Response: Unfortunately, the papers do not provide this information, so we have highlighted this in the discussion

Round 2

Reviewer 1 Report

Many thanks for responding so thoroughly and considerately to previous comments.